# SEMI-DISCRETE NORMALIZING FLOWS THROUGH DIFFERENTIABLE VORONOI TESSELLATION

**Ricky T. Q. Chen, Brandon Amos, Maximilian Nickel**
Facebook AI Research
`rtqichen,bda,maxn@fb.com`

## ABSTRACT

Mapping between discrete and continuous distributions is a difficult task and many have had to resort to approximate or heuristical approaches. We propose a tessellation-based approach that directly learns quantization boundaries on a continuous space, complete with exact likelihood evaluations. This is done through constructing normalizing flows on convex polytopes defined via a differentiable tessellation. Using a simple homeomorphism with an efficient log determinant Jacobian, we can then cheaply parameterize distributions on bounded domains.

We explore this approach in two application settings, mapping from discrete to continuous and vice versa. Firstly, a *Voronoi dequantization* allows automatically learning quantization boundaries in a multidimensional space. The location of boundaries and distances between regions can encode useful structural relations between the quantized discrete values. Secondly, a *Voronoi mixture model* has constant computation cost for likelihood evaluation regardless of the number of mixture components. Empirically, we show improvements over existing methods across a range of structured data modalities, and find that we can achieve a significant gain from just adding Voronoi mixtures to a baseline model.

## 1 INTRODUCTION

Likelihood-based models have seen increasing usage across multiple data modalities. In particular, the family of normalizing flows stands out as a large amount of structure can be incorporated into the model, aiding its usage in modeling a wide variety of domains such as images [7, 22], graphs [28], invariant distributions [24, 2] and molecular structures [40]. However, the majority of works focus on only continuous functions and continuous random variables. This restriction can make it difficult to apply such models to distributions with implicit discrete structures. In this work, we incorporate discrete structure into standard normalizing flows, while being entirely composable with any other normalizing flow. Specifically, we propose a homeomorphism between an unbounded domain and a convex polytope—defined through a learnable tessellation of the domain. This homeomorphism is cheap to compute, has a cheap inverse, and we additionally provide an efficient and exact formulation of the resulting change in density. In other words, this transformation is highly scalable.

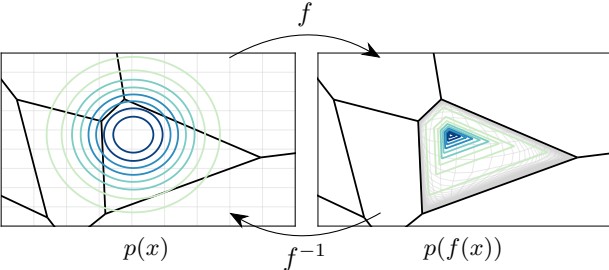

Figure 1: We propose an invertible mapping $f$ between $\mathbb{R}^D$ and a convex polytope, which is parameterized based on a differentiable Voronoi tessellation of $\mathbb{R}^D$. This mapping adds discrete structure into normalizing flows, and its inverse $f^{-1}$ and log determinant Jacobian can both be efficiently computed for high dimensions.

## 2 PRELIMINARIES

**Normalizing Flows.** This family of generative models [34, 23] typically includes any model that makes use of invertible transformations $f$ to map samples between distributions. The relationship between the original and transformed density functions have a closed form expression,

$$p_z(f(x)) = p_x(x) \left| \det \frac{\partial f(x)}{\partial x} \right|^{-1}. \tag{1}$$

If the domain and codomain of $f$ are different, then $p_x$ and $p_z$ can have different supports, *i.e.* regions where probability is non-zero. However, existing designs of $f$ that have this property only act in a single dimension. The use of general support-modifying invertible transformations have not been discussed extensively in the literature.

**Dequantization.** When modeling discrete data with density models, standard approaches typically rely on dequantization methods. These methods provide a correspondence between discrete values and continuous sub-spaces, where a discrete variable $\boldsymbol{y} \in \mathcal{Y}$ is randomly placed within a disjoint subset of $\mathbb{R}^D$ [42].

Let $A_{\boldsymbol{y}}$ denote the subspace corresponding to the value of $\boldsymbol{y}$ and let $q(\boldsymbol{x}|\boldsymbol{y})$ be the dequantization model which has a bounded support in $A_{\boldsymbol{y}}$. Then a density model $p(\boldsymbol{x})$ with support over $\mathbb{R}^D$ satisfies

$$\begin{aligned} \log p(\boldsymbol{y}) &\geq \mathbb{E}_{\boldsymbol{x} \sim q(\boldsymbol{x}|\boldsymbol{y})} \left[ \log(\mathbb{1}_{[\boldsymbol{x} \in A_{\boldsymbol{y}}]} p(\boldsymbol{x})) - \log q(\boldsymbol{x}|\boldsymbol{y}) \right] \\ &= \mathbb{E}_{\boldsymbol{x} \sim q(\boldsymbol{x}|\boldsymbol{y})} \left[ \log p(\boldsymbol{x}) - \log q(\boldsymbol{x}|\boldsymbol{y}) \right] \end{aligned} \tag{2}$$

Thus with an appropriate choice of dequantization, maximizing the likelihood under the density model $p(\boldsymbol{x})$ is equivalent to maximizing the likelihood under the discrete model $p(\boldsymbol{y})$.

To the best of our knowledge, in all existing dequantization methods, the value of $D$ and the choice of subsets $A_{\boldsymbol{y}}$ are entirely dependent on the type of discrete data (ordinal vs non-ordinal) and the number of discrete values $|\mathcal{Y}|$ in the non-ordinal case. Furthermore, the subsets $A_{\boldsymbol{y}}$ do not interact with one another and are fixed during training. In contrast, we conjecture that important properties such as (dis)similarities between discrete values should be modeled as part of the parameterization of $A_{\boldsymbol{y}}$, and that it'd be useful to be able to automatically adapt these subsets $A_{\boldsymbol{y}}$ based on gradients from $p(\boldsymbol{x})$.

**Disjoint mixture models.** Building mixture models is one of the simplest methods for creating more flexible distributions from simpler one. However, in practice it is computationally expensive for large number of mixture components, because evaluating the likelihood under a mixture model requires evaluating the likelihood of each component. To alleviate this issue, Dinh et al. [8] recently proposed to use components with disjoint support. In particular, let $\{A_k\}_{k=1}^K$ be disjoint subsets of $\mathbb{R}^D$, such that each mixture component is defined on one subset and has support restricted to that particular subset. The likelihood of the mixture model then simplifies to

$$p(\boldsymbol{x}) = \sum_{k=1}^K p(\boldsymbol{x}|k)p(k) = \sum_{k=1}^K \mathbb{1}_{[\boldsymbol{x} \in A_k]} p(\boldsymbol{x}|k)p(k) = p(\boldsymbol{x}|k = g(\boldsymbol{x}))p(k = g(\boldsymbol{x})) \tag{3}$$

where $g : \mathbb{R}^D \to \{1, \ldots, K\}$ is a set identification function that satisfies $\boldsymbol{x} \in A_{g(\boldsymbol{x})}$. This framework allows building large mixture models while no longer having a compute cost that scales with $K$. In contrast to variational approaches with discrete latent variables, the use of disjoint subsets provides an exact log-likelihood and not a lower bound.

## 3 VORONOI TESSELLATION FOR NORMALIZING FLOWS

We separate the domain into subsets through a Voronoi tessellation [46]. This induces a correspondence between each subset and a corresponding *anchor point*, which provides a parameterization of the tessellation for gradient-based optimization.

Let $X = \{x_1, \ldots, x_K\}$ be a set of anchor points in $\mathbb{R}^D$. The Voronoi cell for each anchor point is given by

$$V_k \triangleq \{x \in \mathbb{R}^D : \|x_k - x\| < \|x_j - x\|, j = 1, \ldots, K\}, \tag{4}$$

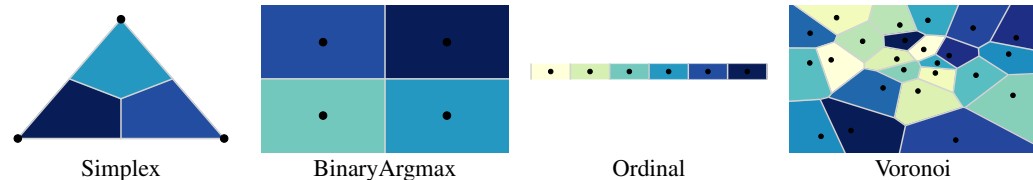

Figure 2: Existing dequantization methods can be seen as special cases of Voronoi dequantization with fixed anchor points. In additional to decoupling the dimension of the embedding space from the number of discrete values, the Voronoi dequantization can learn to model the similarities of discrete values through the positions and boundaries of their Voronoi cells.

*i.e.* it defines a subspace containing all points which have the anchor points as its nearest neighbor. This subspace can equivalently be expressed in the form of a convex polytope,

$$V_k = \{x \in \mathbb{R}^D : Ax < b\},$$
$$\text{where } A_i = 2(x_i - x_k)^\mathsf{T}, b_i = \|x_i\|^2 - \|x_k\|^2. \tag{5}$$

For simplicity, we also include box constraints so that all Voronoi cells are bounded in all directions.

$$V_k = \{x \in \mathbb{R}^D : Ax < b, c_l < x < c_r\} \tag{6}$$

Thus, the learnable parameters of this Voronoi tessellation are the anchor points $\{x_1, \ldots, x_K\}$, and the box constraints $c_l, c_r \in \mathbb{R}^D$.

## 3.1 INVERTIBLE MAPPING ONTO THE VORONOI CELL

Given a cell $V_k$ for some $k \in \{1, \ldots, K\}$, we construct an invertible mapping $f_k : \mathbb{R}^D \to V_k$ by the following 2-step procedure.

Let $\boldsymbol{x} \in \mathbb{R}^D$. If $\boldsymbol{x} = \boldsymbol{x_k}$, the anchor point of $V_k$, we simply set $f_k(\boldsymbol{x}) = \boldsymbol{x}$. Otherwise, the first step is as follows:

1. Find where the ray that starts from $\boldsymbol{x_k}$ in the direction of $\boldsymbol{x}$ intersects with the boundary of $V_k$.

Since $V_k$ is a convex polytope, we can frame this as a linear programming problem. First define the direction $\boldsymbol{\delta_k}(\boldsymbol{x}) \triangleq \frac{\boldsymbol{x} - \boldsymbol{x_k}}{\|\boldsymbol{x} - \boldsymbol{x_k}\|}$ and the ray $\boldsymbol{x}(\lambda) \triangleq \boldsymbol{x_k} + \lambda \boldsymbol{\delta_k}(\boldsymbol{x})$, with $\lambda > 0$. Then this problem can be solved through the following linear program:

$$\max \lambda \quad \text{s.t.} \quad \boldsymbol{x}(\lambda) \in \overline{V_k}, \lambda \geq 0 \tag{7}$$

Let $\lambda^*$ be the solution. This first step solves for the farthest point in the Voronoi cell in the direction of $x$. Using this knowledge, we can now map all points that lie on this ray onto the Voronoi cell.

1. Apply an invertible transformation such that any point on the ray $\{\boldsymbol{x}(\lambda) : \lambda > 0\}$ is mapped onto the line segment $\{\boldsymbol{x_k} + \alpha(\boldsymbol{x}(\lambda^*) - \boldsymbol{x_k}) : \alpha \in (0, 1)\}$.

There are many possible choices for designing this transformation. An intuitive choice is to use a monotonic transformation of the distance from $\boldsymbol{x}$ to the anchor point $\boldsymbol{x_k}$.

$$f_k(\boldsymbol{x}) \triangleq \boldsymbol{x_k} + \alpha_k \left( \frac{\|\boldsymbol{x} - \boldsymbol{x_k}\|}{\|\boldsymbol{x}(\lambda^*) - \boldsymbol{x_k}\|} \right) (\boldsymbol{x}(\lambda^*) - \boldsymbol{x_k}) \tag{8}$$

where $\alpha_k$ is an appropriate *invertible* squashing function from $R^+$ to $(0, 1)$. In our experiments, we use $\alpha_k(h) = \text{softsign}(\gamma_k h)$ where $\gamma_k$ is a learned cell-dependent scale.

**Remarks** The problem in Equation (7) can be solved exactly and without the use of an iterative convex optimization solver, which also provides analytical derivatives of the solution. The inverse computation shares the same first step of this procedure. Finally, we provide a method of computing the exact log determinant Jacobian that only involves dot products, thus efficiently scaling with the number of dimensions. See detailed explanations of these remarks and propositions in Appendix A.

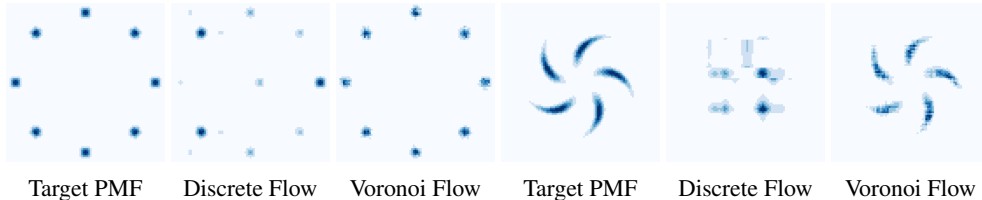

| Target PMF | Discrete Flow | Voronoi Flow | Target PMF | Discrete Flow | Voronoi Flow |

Figure 3: **2D quantized toy data.** Voronoi Flows can model complex relations between discrete values. Each of the two discrete variables have 91 possible values, and no knowledge of ordering is given to the models.

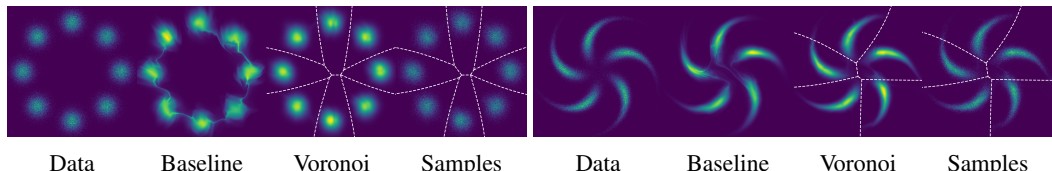

| Data | Baseline | Voronoi | Samples | Data | Baseline | Voronoi | Samples |

Figure 4: Tessellation can be done in a transformed space; nonlinear boundaries are shown.

**Applications**   We apply this approach to dequantization—which maps from discrete variables to continuous variables—and disjoint mixture modeling—which maps from continuous to discrete. Importantly, all existing dequantization boundaries are special cases of Voronoi tessellations (see Figure 2). Our approach allows us to directly parameterize and learn these boundaries through just gradient-based optimization. Detailed explanations are in Appendix B.

## 4   EXPERIMENTS

Additional experiments regarding permutation-invariant itemset modeling and language modeling are in Appendix D.

**Voronoi dequantization**   We start by considering synthetic distribution over two discrete variables, to compare against Discrete Flows [43]. Figure 3 shows the target probability mass functions, which are created by a 2D distribution where each dimension is quantized into 91 discrete values. Though the data is ordinal, no knowledge of ordering is given to the models. We see that Discrete Flows has trouble fitting these, presumably because it is difficult to rearrange the target distribution into a factorized distribution. We also show results on UCI discrete data sets in Table 2.

**Voronoi mixture models**   We trained disjoint mixture models using the inverse of our transformation, where data is mapped from each bounded Voronoi cell onto the unbounded space, which is then composed with a conditional normalizing flow model. Results are shown in Section 3.1. The boundaries are nonlinear as it is done after a nonlinear transformation.

| Method | Connect4 | Forests | Mushroom | Nursery | PokerHands | USCensus90 |
|---|---|---|---|---|---|---|
| Voronoi Deq. | $12.92_{\pm 0.07}$ | $14.20_{\pm 0.05}$ | $9.06_{\pm 0.05}$ | $9.27_{\pm 0.04}$ | $19.86_{\pm 0.04}$ | $24.19_{\pm 0.12}$ |
| Simplex Deq. | $13.46_{\pm 0.01}$ | $16.58_{\pm 0.01}$ | $9.26_{\pm 0.01}$ | $9.50_{\pm 0.00}$ | $19.90_{\pm 0.00}$ | $28.09_{\pm 0.08}$ |
| BinaryArgmax Deq. | $13.71_{\pm 0.04}$ | $16.73_{\pm 0.17}$ | $9.53_{\pm 0.01}$ | $9.49_{\pm 0.00}$ | $19.90_{\pm 0.01}$ | $27.23_{\pm 0.02}$ |
| Discrete Flow | $19.80_{\pm 0.01}$ | $21.91_{\pm 0.01}$ | $22.06_{\pm 0.01}$ | $9.53_{\pm 0.01}$ | $19.82_{\pm 0.03}$ | $55.62_{\pm 0.35}$ |

Table 1: **Discrete UCI data sets.** Negative log-likelihood results on the test sets in nats.

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

## A    ADDITIONAL REMARKS AND PROPOSITIONS

**Box constraints.**    There can be continuity problems if a Voronoi cell is unbounded, as the solution to Equation (8) does not exist if $x(\lambda^*)$ diverges. Furthermore, when solving Equation (7) it can be difficult to numerically differentiate between an unbounded cell and one whose boundaries are very far away. It is for these reasons that we introduce box constraints (Equation 6) in the formulation of Voronoi cells which allows us to sidestep these issues for now.

**Solving for $\lambda^*$.**    While Equation (7) can be solved using a convex optimization library, this approach is prone to numerical errors and requires differentiating through convex optimization solutions for gradient-based learning [1]. We instead note that the solution of Equation (7) can be expressed in closed form, since it is always going to be the intersection of the ray $x(\lambda)$ and a linear constraint.

Let $a_i^\mathsf{T} x = b_i$ be the plane that represents one of the linear constraints in Equation (6). Let $\lambda_i$ be the intersection of this plane with the ray, *i.e.* it is the solution to $a_i^\mathsf{T} x(\lambda_i) = b_i$. This can be solved exactly as

$$\lambda_i = \frac{b_i - a_i^\mathsf{T} x_k}{a_i^\mathsf{T} \delta_k(x)}. \tag{9}$$

Then the solution is simply the smallest positive $\lambda_i$, which satisfies all the linear constraints, $\lambda^* = \min\{\lambda_i : \lambda_i > 0\}$. There are a total of $K + 2D - 1$ linear constraints, including the Voronoi cell boundaries and box constraints, which can be computed fully in parallel.

**Automatic differentiation.**    Note that the above formulation of the solution $\lambda^*$ also provides gradients $\frac{d\lambda^*}{d\delta_k(x)}$ through automatic differentiation, since it only involves primitive operations. This allows end-to-end differentiation of the mapping $f_k$, and will allow us to learn the parameters of $f_k$, *i.e.* parameters of $\alpha_k$ and the Voronoi tessellation.

**Proposition 1.** *The mapping $f_k : \mathbb{R}^D \to V_k$ as defined in the 2-step procedure is a homeomorphism.*

That is to say, $f_k$ is a bijection, and both $f_k$ and $f_k^{-1}$ are continuous. This allows us to use $f_k$ within the normalizing flows framework, as a mapping between a distribution $p_x$ defined on $R^D$ and the transformed distribution $p_z$ on $V_k$, with change in density given by

$$p_z(f_k(x)) = p_x(x) \left| \det \frac{df_k(x)}{dx} \right|^{-1}. \tag{10}$$

**Proposition 2.** *If $p_x(x)$ is continuous, then the transformed density $p_z(f_k(x))$ is continuous almost everywhere.*

This comes from the Jacobian being continuous a.e.

### A.1    COMPUTING THE INVERSE MAPPING

We will also be using the inverse mapping $f_k^{-1} : V_k \to \mathbb{R}^D$, so we next describe how to compute this. Let $z = f_k(x)$.

Conveniently, since both $x$ and $z$ lie on the ray $\{x(\lambda) : \lambda > 0\}$, we know $\delta_k(x) = \delta_k(z)$. So the first step is the same as the forward procedure: we solve for $\lambda^*$ and $x(\lambda^*)$. Following this, we then recover $x$ by inverting Step 2 of the forward procedure.

This inverse transformation is given by

$$\tilde{\alpha} = \frac{z - x_k}{x(\lambda^*) - x_k} \tag{11}$$

$$\tilde{\Delta} = \alpha_k^{-1}(\tilde{\alpha}_1) \tag{12}$$

$$\Delta = \tilde{\Delta} \, \|x(\lambda^*) - x_k\| \tag{13}$$

$$x = \Delta \delta_k(z) + x_k \tag{14}$$

Equation (11) is an element-wise division. Since $\tilde{\alpha}$ will the same in all dimensions, we can simply pick a dimension in Equation (12). In our experiments, the inverse $\alpha_k^{-1}$ can be computed analytically, though since it is just a scalar function, simple methods like bisection can also work when the inverse is not known analytically. Lastly, Equation (14) follows from the observation that $\boldsymbol{\delta_k(x)} = \boldsymbol{\delta_k(z)}$.

### A.2 EFFICIENT COMPUTATION OF THE LOG DET JACOBIAN

As $f_k$ is a mapping in $D$ dimensions, computing the log determinant Jacobian for likelihood computation can be costly and will scale poorly with $D$ if computed naïvely. Instead, we note that the Jacobian of $f_k$ is highly structured.

Intuitively, because $f_k$ depends only on the direction $\boldsymbol{\delta_k(x)}$ and the distance away from $\boldsymbol{x_k}$, it only has two degrees of freedom regardless of $D$. In fact, the Jacobian of $f_k$ can be represented as a rank-2 update on a scaled identity matrix. This allows us to use the matrix determinant lemma to reformulate the log determinant Jacobian in a compute- and memory-efficient form. We summarize this in a proposition.

**Proposition 3.** *Let the transformation $f_k(x)$ and all intermediate quantities be as defined in Section 3.1 for some given input $x$. Then the Jacobian factorizes as*

$$\frac{\partial f_k(x)}{\partial x} = cI + u_1 v_1^{\mathsf{T}} + u_2 v_2^{\mathsf{T}} \tag{15}$$

*for some $c \in \mathbb{R}, u_i \in \mathbb{R}^D, v_i \in \mathbb{R}^D$, and its log determinant has the form*

$$
\begin{aligned}
\log &\left| \det \frac{\partial f_k(x)}{\partial x} \right| \\
&= \log |1 + w_{11}| + \log \left| 1 + w_{22} - \frac{w_{12} w_{21}}{1 + w_{11}} \right| + D \log c
\end{aligned}
\tag{16}
$$

where $w_{ij} = c^{-1} v_i^{\mathsf{T}} u_j$. This expression for the log determinant only requires dot products between vectors in $R^D$. To reduce notational clutter, the exact formulas for $c, u_1, u_2, v_1, v_2$ can be found in Appendix F.

**Remark on computational cost.** All vectors used in computing Equation (16) are either readily available as intermediate quantities after computing $f_k(x)$, or are gradients of scalar functions and can be efficiently computed through reverse-mode automatic differentiation. The only operations on these vectors are dot products, and no large $D$-by-$D$ matrices are ever constructed as part of any intermediate steps. Compared to explicitly constructing the Jacobian matrix, this is more efficient in both compute and memory and can readily scale to large values of $D$.

## B APPLICATION SETTINGS

We discuss two applications of our method to likelihood-based modeling of discrete and continuous data.

The first is a novel dequantization method that allows training a model of discrete data using density models that normally only act on continuous variables such as normalizing flows. Compared to existing dequantization methods [7, 36, 17], the Voronoi dequantization is not restricted to any fixed-dimensional continuous space and can benefit from learning similarities between discrete values. In fact, previous approaches can be seen as special cases of a Voronoi dequantization with fixed equidistant anchor points.

The second is a novel formulation of disjoint mixture models, where each Voronoi cell represents a single component in a large mixture model. To the best of our knowledge, disjoint mixture models have not been explored significantly in the literature and have not been successfully applied in more than a couple dimensions. Compared to an existing method [8], the Voronoi mixture model is not restricted to acting individually for each dimension and we empirically show can scale to thousands of dimensions.

### B.1 VORONOI DEQUANTIZATION

Let $\boldsymbol{y}$ be a discrete random variable with finite support $\mathcal{Y}$. Then we can use Voronoi tessellation to define subsets for dequantizing $\boldsymbol{y}$ as long as there are at least as many Voronoi cells—equivalently, anchors points—as the number of discrete values, *i.e.* $K \geq |\mathcal{Y}|$. By assigning each discrete value to a Voronoi cell, we can then define a dequantization model $q(\boldsymbol{x}|\boldsymbol{y})$ by first sampling from a base (*e.g.* Gaussian) distribution $\boldsymbol{z} \sim q(\boldsymbol{z}|\boldsymbol{y})$ in $\mathbb{R}^D$ and then applying the mapping $\boldsymbol{x} = f_k(\boldsymbol{z})$ from Section 3.1 to construct a distribution over $V_k$. We can then obtain probabilities $q(\boldsymbol{x}|\boldsymbol{y})$ efficiently using Proposition 3 and train the dequantization alongside the density model $p(\boldsymbol{x})$ on $\mathbb{R}^D$.

Automatic differentiation through $p(\boldsymbol{x})$ and $f_k$ (as discussed in Appendix A) provides gradients for training $q(\boldsymbol{x}|\boldsymbol{y})$, which include gradient with respect to parameters of the Voronoi tessellation. We are also free to choose the number of dimensions $D$, where a smaller $D$ induces dequantized distributions that may be easier to fit while a larger $D$ allows the anchor points and Voronoi cells more room to change over the course of training.

Sampling from the model $p(\boldsymbol{y}|\boldsymbol{x})$ is straightforward and deterministic after sampling $\boldsymbol{x}$. We can write $\boldsymbol{y} = g(\boldsymbol{x})$ where $g$ is the set identification function satisfying $\boldsymbol{x} \in V_{g(\boldsymbol{x})}$. From the definition of Voronoi tessellation Equation (4), it is easy to see that $g(\boldsymbol{x})$ is the nearest neighbor operation,

$$g(\boldsymbol{x}) = \arg \min_k \|\boldsymbol{x} - \boldsymbol{x_k}\|. \tag{17}$$

Depending on the position of anchor points, we can recover the disjoint subsets used by prior methods as special cases. We illustrate this in Figure 2.

### B.2 VORONOI MIXTURE MODELS

Let $\{V_k\}$ be a Voronoi tessellation of $\mathbb{R}^D$. Then a disjoint mixture model can be constructed by defining distributions on each Voronoi cell. Here we make use of the inverse mapping $f_k^{-1} : V_k \to \mathbb{R}^D$ so that we only need to parameterize distributions over $\mathbb{R}^D$. Let $\boldsymbol{x}$ be a point in $\mathbb{R}^D$, our Voronoi mixture model assigns the density

$$\begin{aligned}
\log p_{\text{mix}}(\boldsymbol{x}) = {} & \log p_{\text{comp}}(f_{k=g(\boldsymbol{x})}^{-1}(\boldsymbol{x})|k = g(\boldsymbol{x})) \\
& + \log \left| \det \frac{\partial f_k^{-1}(\boldsymbol{x})}{\partial \boldsymbol{x}} \right| \\
& + \log p(k = g(\boldsymbol{x}))
\end{aligned} \tag{18}$$

where $p_{\text{comp}}$ can be any distribution over $\mathbb{R}^D$.

A Voronoi mixture model can also be viewed as a type of normalizing flow where, in addition to the change of variable due to $f_k^{-1}$, we also need to apply the change in density resulting from choosing one out of $K$ components, $p(k)$. Per the observation of Dinh et al. [8], we have transformed $\boldsymbol{x}$ into a tuple of a continuous variable $\boldsymbol{z} = f_k^{-1}(\boldsymbol{x})$ and a discrete auxiliary variable $k$. A natural extension is to repeatedly apply this transformation, effectively forming a deep mixture model [41, 45], and resulting in a sequence of discrete auxiliary variables. This lets us define large mixture models hierarchically without the need to create a tessellation with a large number of Voronoi cells. Crucially, even though the number of mixture components increases exponentially with respect to depth, the computational cost does not scale with the number of components.

We also provide a simple proposition that the resulting density function is amenable to gradient descent.

**Proposition 4.** *For any distribution $p(k)$ with support over $\{1, \ldots, K\}$, define the mixture distribution,*

$$p(\boldsymbol{x}) = \sum_{k=1}^{K} p(\boldsymbol{x}|k)p(k), \tag{19}$$

*where $p(\cdot|k)$ is the distribution mapped onto the Voronoi cell $V_k$ (i.e. Equation 10) from a density that is continuous. Then the density function of the mixture is continuous a.e.*

Unlike Dinh et al. [8], we don't need to worry about the density being discontinuous for certain parameterizations.

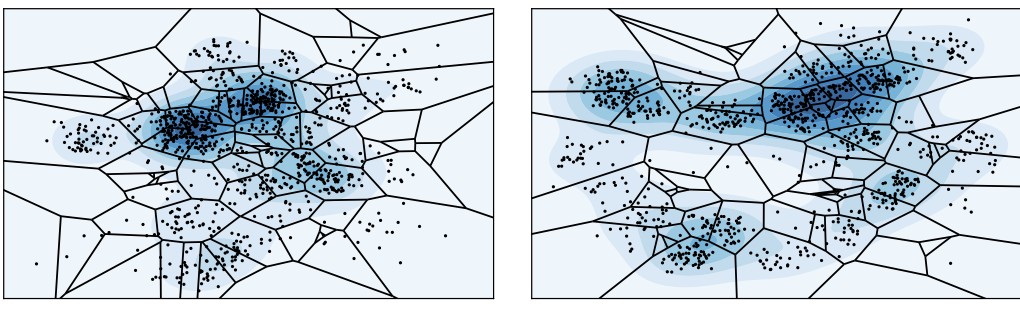

Dequantized samples for $\boldsymbol{y}_0$        Dequantized samples for $\boldsymbol{y}_1$

Figure 5: Dequantized samples from model trained on quantized `8gaussians` data set. Each plot shows samples from a single discrete variable, mapped onto a Voronoi tessellation of $\mathbb{R}^2$, along with a kernel density estimator. The model learns to cluster discrete values with similar probability values.

## C    RELATED WORK

**Normalizing flows for discrete data.** Invertible mappings have been proposed for discrete data, where discrete values are effectively rearranged from a factorized distribution. In order to parameterize the transformation, Hoogeboom et al. [16], van den Berg et al. [44] use quantized ordinal transformations, while Tran et al. [43] takes a more general approach of doing modulo addition on one-hot vectors. These approaches suffer from gradient bias due to the need to use discontinuous operations and do not have universality guarantees since it's unclear whether simple rearrangement is sufficient to transform any joint discrete distribution into a factorized distribution. In contrast, the dequantization approach provides a universality guarantee since the lower bound in Equation (2) is tight when $p(\boldsymbol{x})\mathbb{1}_{[\boldsymbol{x} \in A_{\boldsymbol{y}}]} \propto q(\boldsymbol{x}|\boldsymbol{y})$, with a proportionality equal to $p(\boldsymbol{y})$.

**Dequantization methods.** Within the normalizing flows literature, the act of adding noise was originally used for ordinal data as a way to combat numerical issues [7]. Later on, appropriate dequantization approaches have been shown to lower bound the log-likelihood of a discrete model [42, 15]. For non-ordinal data, many works have proposed simplex-based approaches. Early works on relaxations [19, 31] proposed continuous distribution on the simplex that mimic the behaviors of discrete random variables; however, these were only designed for the use with a Gumbel base distribution. Potapczynski et al. [36] extend this to a Gaussian distribution—although it is not hard to see this can work with any base distribution—by designing invertible transformations between $R^D$ and the probability simplex with $K$ vertices, with $D = K - 1$, where $K$ is the number of classes of a discrete random variable.

Intuitively, after fixing one of the logits, the softmax operation is an invertible transformation from $\mathbb{R}^{K-1}$ to $\{\boldsymbol{x} \in \mathbb{R}^K : \sum_i^K \boldsymbol{x}_i = 1, \boldsymbol{x}_i > 0\}$. The $(K-1)$-simplex can then be broken into $K$ subsets, each corresponding to a particular discrete value.

$$A_k = \{\boldsymbol{x} \in \mathbb{R}^K : \textstyle\sum_{i=1}^K \boldsymbol{x}_i = 1, \boldsymbol{x}_k > \boldsymbol{x}_i \; \forall i \neq k\}. \tag{20}$$

More recently, Hoogeboom et al. [17] proposed ignoring the simplex constraint and simply use

$$A_k = \{\boldsymbol{x} \in \mathbb{R}^K : \boldsymbol{x}_k > \boldsymbol{x}_i \; \forall i \neq k\}, \tag{21}$$

which effectively increases the number of dimensions by one compared to the simplex approach. However, both simplex-based approaches force the dimension of the continuous space $D$ to scale with $K$. In order to make simplex-based dequantization work when $K$ is large, they propose reducing all discrete variables to a set of binary variables before applying dequantization. In contrast, Voronoi dequantization has full flexibility is choosing $D$ regardless of $K$.

A number of works [30, 27] also proposed removing the constraint that subsets are disjoint, and instead work with general mixture models with unbounded support, relying on the conditional model $p(\boldsymbol{y}|\boldsymbol{x})$ being sufficiently weak so that the task of modeling is forced onto a flow-based prior. Similar to general approaches that combine normalizing flows with variational inference approaches [48, 18, 32],

| Method | Connect4 | Forests | Mushroom | Nursery | PokerHands | USCensus90 |
|---|---|---|---|---|---|---|
| Voronoi Deq. | $12.92_{\pm 0.07}$ | $14.20_{\pm 0.05}$ | $9.06_{\pm 0.05}$ | $9.27_{\pm 0.04}$ | $19.86_{\pm 0.04}$ | $24.19_{\pm 0.12}$ |
| Simplex Deq. | $13.46_{\pm 0.01}$ | $16.58_{\pm 0.01}$ | $9.26_{\pm 0.01}$ | $9.50_{\pm 0.00}$ | $19.90_{\pm 0.00}$ | $28.09_{\pm 0.08}$ |
| BinaryArgmax Deq. | $13.71_{\pm 0.04}$ | $16.73_{\pm 0.17}$ | $9.53_{\pm 0.01}$ | $9.49_{\pm 0.00}$ | $19.90_{\pm 0.01}$ | $27.23_{\pm 0.02}$ |
| Discrete Flow | $19.80_{\pm 0.01}$ | $21.91_{\pm 0.01}$ | $22.06_{\pm 0.01}$ | $9.53_{\pm 0.01}$ | $19.82_{\pm 0.03}$ | $55.62_{\pm 0.35}$ |

Table 2: **Discrete UCI data sets.** Negative log-likelihood results on the test sets in nats.

they have achieved good performance on a number of tasks. However, they lose the computational savings and the deterministic decoder $p(\boldsymbol{y}|\boldsymbol{x})$ gained from using disjoint subsets. On the other hand, quantization based on nearest neighbor have been for learning discrete latent variables [33, 38], but no likelihoods are constructed, the boundaries are not explicitly differentiated through, and the model relies on training with modified gradients based on heuristics.

**Disjoint mixture models.** The computational savings from using disjoint subsets was pointed out by Dinh et al. [8]. However, their method only works in each dimension individually. They transform samples using a linear spline, which is equivalent to creating subsets based on the knots of the spline and apply a linear transformation within each subset. Furthermore, certain parameterizations of the spline can lead to discontinuous density functions, whereas our disjoint mixture has a density function that is continuous almost everywhere. Overall, splines are interesting because each segment is independent, and the use of monotonic splines have been combined with normalizing flows [10]. However, the splines in Dinh et al. [8] are not enforced to be monotonic, so the full transformation is not bijective and acts like a disjoint mixture model. Ultimately, their experiments were restricted to two-dimensional syntheic data sets, and it remained an interesting research question whether disjoint mixture models can be successfully applied in high dimensions.

## D  EXPERIMENTS

We experimentally validate our semi-discrete approach of combining Voronoi tessellation within likelihood-based modeling on a variety of data domains: discrete-valued UCI data sets, itemset modeling, language modeling, and disjoint mixture modeling. The goal of these experiments is not to show state-of-the-art results in these domains but to showcase the relative merit of our use of Voronoi tessellation compared to existing methods. For this reason, we just use basic coupling blocks with affine transformations [6, 7] as a base flow model for our experiments, unless stated otherwise. When comparing to Discrete Flows [43], we use their bipartite layers. Details regarding preprocessing for data sets can be found in Appendix G, and detailed experimental setup is in Appendix H.

### D.1  2D SYNTHETIC DATA

We start by considering synthetic distribution over two discrete variables, to compare against Discrete Flows [43]. Figure 3 shows the target probability mass functions, which are created by a 2D distribution where each dimension is quantized into 91 discrete values. Though the data is ordinal, no knowledge of ordering is given to the models. We see that Discrete Flows has trouble fitting these, presumably because it is difficult to rearrange the target distribution into a factorized distribution.

For our model, we dequantize each discrete variable with a Voronoi tessellation in $\mathbb{R}^2$. We then learn a flow model on the combined $\mathbb{R}^4$, parameterized by multilayer perceptrons (MLPs). In Figure 5 we visualize the learned Voronoi tessellation and samples from our model. The learned tessellation seems to group some of discrete values that occur frequently together, so the resulting model can have less modes.

### D.2  DISCRETE-VALUED UCI DATA

We experiment with complex data sets where each discrete variable can have a varying number of classes. Furthermore, these discrete variables may have hidden semantics. To this end, we use

| Model | Dequantization | Retail | Accidents |
|---|---|---|---|
| Equivariant CNF | Voronoi | $9.44_{\pm 2.34}$ | $7.81_{\pm 2.84}$ |
| Equivariant CNF | Simplex | $24.16_{\pm 0.21}$ | $19.19_{\pm 0.01}$ |
| Equivariant CNF | BinaryArgmax | $10.47_{\pm 0.42}$ | $6.72_{\pm 0.23}$ |
| Determinantal Point Process | | $20.35_{\pm 0.05}$ | $15.78_{\pm 0.04}$ |

Table 3: **Permutation-invariant itemset modeling.** Negative log-likelihood results on the test sets in nats.

| Dequantization | text8 | enwik8 |
|---|---|---|
| Voronoi ($D$=2) | $1.39_{\pm 0.01}$ | $1.46_{\pm 0.01}$ |
| Voronoi ($D$=4) | $1.37_{\pm 0.00}$ | $1.41_{\pm 0.00}$ |
| Voronoi ($D$=6) | $1.37_{\pm 0.00}$ | $1.40_{\pm 0.00}$ |
| Voronoi ($D$=8) | $1.36_{\pm 0.00}$ | $1.39_{\pm 0.01}$ |
| Argmax | 1.38 | 1.42 |

Table 4: **Character-level language modeling.** Negative log-likelihood results on the test sets in bits per character.

unprocessed data sets from the UCI database [9]. The only pre-processing we perform is to remove variables that only have one discrete value. We then take 80% as train, 10% as validation, and 10% as the test set. Most of these datasets have a combination of both ordinal and non-ordinal variables, and we expect the non-ordinal variables to exhibit relations that are unknown (*e.g.* spatial correlations).

We see that Discrete Flows can be competitive with dequantization approaches, but can also fall short on difficult data sets such as the `USCensus90`, the largest data set we consider with 2.7 million examples and 68 different discrete variables of varying types. For dequantization methods, simplex [36] and binary argmax [17] approaches are mostly on par. We do see a non-negligible gap in performance between these baselines and Voronoi dequantization for most of the data sets, likely due to the ability to learn semantically useful relations between the values of each discrete variable. For instance, the `Connect4` dataset contains positions of a two-player board game with the same name, which exhibit complex spatial dependencies between the board pieces, and the `USCensus90` dataset contains highly correlated variables and is often used to test clustering algorithms.

### D.3 PERMUTATION-INVARIANT ITEMSET MODELING

A major appeal of using normalizing flows in continuous space is the ability to incorporate invariance to specific group symmetries into the density model. For instance, this can be done by ensuring the ordinary differential equation is equivariant [24, 2, 40] in a continuous normalizing flow [5, 12] framework. Here we focus on invariance with respect to permutations, *i.e.* sets of discrete variables. This invariance cannot be explicitly modeled by Discrete Flows as they require an ordering or a bipartition of discrete variables.

We preprocessed a data set of retail market basket data from an anonymous Belgian retail store [3] and a data set of anonymized traffic accident data [11], which contain sets of discrete variables each with 765 and 269 values, respectively for each data set. Given the large number of discrete values, we expected Voronoi dequantization to perform better. However, we see that in Table 4, binary argmax is quite competitive, likely because it decomposes the full distribution using multiple binary variables. We did not do this binarization trick for simplex dequantization, and we see that it suffers from having to use large embedding dimensions, performing worse than a determinental point process baseline [25].

### D.4 Language modeling

As a widely used benchmark for discrete models [43, 27, 17], we also experiment with our method on language modeling. Here we used the open source code provided by Hoogeboom et al. [17] with the exact same autoregressive flow model and optimizer setups. The only difference is replacing their binary argmax with our Voronoi dequantization. Results are shown in Table 4, where we tried out multiple embedding dimensions $D$. Generally, we find $D = 2$ to be too low and can stagnate training since the Voronoi cells are more constrained. The flexibility in choosing $D$ gives us a slight edge in performance.

## E   Conclusion and Discussion

We combine Voronoi tessellation with normalizing flows to construct a new invertible transformation that has learnable discrete structure. This acts as a learnable mapping between discrete and continuous distributions. We propose two applications of this method: a Voronoi dequantization that maps discrete values into a learnable convex polytope, and a Voronoi mixture modeling approach that has the same compute cost as a single component. We showcased the relative merit of our approach across a range of data modalities with varying structure.

### E.1   Limitations and Future Directions

**Diminishing density on boundaries.**   The distributions within each Voronoi cell necessarily go to zero on the boundary between cells due to the use of a homeomorphism from an unbounded domain. This is a different and undesirable property as opposed to the partitioning method used by [8]. However, alternatives could require more compute cost in the form of solving normalization constants. Balancing expressiveness and compute cost is a delicate problem and sits at the core of tractable probabilistic modeling.

**Design of homeomorphisms and tessellations.**   Our proposed homeomorphism is simple and scalable, but this comes at the cost of smoothness and expressiveness. As depicted in Figure 1, the transformed density has a non-smooth probability density function. This non-smoothness exists when the ray intersections with multiple boundaries. This may cause optimization to be difficult as the gradients of the log-likelihood objective can be discontinuous. Additionally, our use of Euclidean distance can become problematic in high dimensions, as this metric can cause almost all points to be nearly equidistant, resulting in a behavior where all points lie seemingly very close to the boundary. Improvements on the design of homeomorphisms to bounded domains could help alleviate these problems. Additionally, more flexible tessellations—such as the Laguerre tessellation—and additional concepts from semi-discrete optimal transport [35, 26, 13] may be adapted to improve semi-discrete normalizing flows.

## F   Proofs

**Proposition 3.** *Let the transformation $f_k(x)$ and all intermediate quantities be as defined in Section 3.1 for some given input $x$. Then the Jacobian factorizes as*

$$\frac{\partial f_k(x)}{\partial x} = cI + u_1 v_1^\mathsf{T} + u_2 v_2^\mathsf{T} \tag{15}$$

*for some $c \in \mathbb{R}, u_i \in \mathbb{R}^D, v_i \in \mathbb{R}^D$, and its log determinant has the form*

$$\log \left| \det \frac{\partial f_k(x)}{\partial x} \right|$$
$$= \log |1 + w_{11}| + \log \left| 1 + w_{22} - \frac{w_{12} w_{21}}{1 + w_{11}} \right| + D \log c \tag{16}$$

*where*

$$\Delta \triangleq \|x - x_k\| \tag{22}$$

$$c \triangleq \alpha_k(\tilde{\Delta})\lambda^* \Delta^{-1} \tag{23}$$

$$u_1 \triangleq \left[ \alpha_k(\tilde{\Delta})\Delta^{-1} - \frac{\partial \alpha_k(\tilde{\Delta})}{\partial \tilde{\Delta}} \right] \boldsymbol{\delta_k(x)} \tag{24}$$

$$v_1 \triangleq \frac{\partial \lambda^*}{\partial \boldsymbol{\delta_k(x)}} \tag{25}$$

$$u_2 \triangleq \left[ \left( \frac{\partial \alpha_k(\tilde{\Delta})}{\partial \tilde{\Delta}} \right) - \alpha_k(\tilde{\Delta})\lambda^* \Delta^{-1} - \alpha_k(\tilde{\Delta})\Delta^{-1} \left( \left( \frac{\partial \lambda^*}{\partial \boldsymbol{\delta_k(x)}} \right)^{\mathsf{T}} \boldsymbol{\delta_k(x)} \right) \right. \tag{26}$$

$$\left. + \left( \frac{\partial \alpha_k(\tilde{\Delta})}{\partial \tilde{\Delta}} \right) \left( \left( \frac{\partial \lambda^*}{\partial \boldsymbol{\delta_k(x)}} \right)^{\mathsf{T}} \boldsymbol{\delta_k(x)} \right) \right] \boldsymbol{\delta_k(x)} \tag{27}$$

$$v_2 \triangleq \boldsymbol{\delta_k(x)} \tag{28}$$

$$w_{ij} \triangleq c^{-1} v_i^{\mathsf{T}} u_j, \text{ for } i, j \in \{1, 2\} \tag{29}$$

*Proof.* We first write the Jacobian of $f_k$ in the form of $cI + u_1 v_1^{\mathsf{T}} + u_2 v_2^{\mathsf{T}}$ where $c$ is a scalar, and $u_1, u_2, v_1, v_2$ are vectors of size $D$. Define the shorthand $\Delta \triangleq \|x - x_k\|$, $\Delta^* \triangleq \|x(\lambda^*) - x_k\|$, and $\tilde{\Delta} \triangleq \Delta/\Delta^*$. To simplify notation, we use the short-hands $\alpha_k = \alpha_k(\tilde{\Delta})$, $\delta_k = \delta_k(x)$. Then the Jacobian follows

$$\frac{\partial f_k(x)}{\partial x} = \frac{\partial}{\partial x} (x_k + \alpha_k(x(\lambda^*) - x_k)) \tag{30}$$

$$= (x(\lambda^*) - x_k) \left( \frac{\partial \alpha_k}{\partial \tilde{\Delta}} \right) \left( \frac{\partial \tilde{\Delta}}{\partial x} \right)^{\mathsf{T}} + \alpha_k \left( \frac{\partial x(\lambda^*)}{\partial x} \right) \tag{31}$$

$$= (x(\lambda^*) - x_k) \left( \frac{\partial \alpha_k}{\partial \tilde{\Delta}} \right) \left( \frac{1}{\Delta^*} \delta_k^{\mathsf{T}} - \frac{\Delta}{(\Delta^*)^2} (x(\lambda^*) - x_k)^{\mathsf{T}} \frac{\partial x(\lambda^*)}{\partial x} \right) + \alpha_k \left( \frac{\partial x(\lambda^*)}{\partial x} \right) \tag{32}$$

$$= \left( \frac{\partial \alpha_k}{\partial \tilde{\Delta}} \right) \delta_k \delta_k^{\mathsf{T}} - \left( \frac{\partial \alpha_k}{\partial \tilde{\Delta}} \right) \Delta \delta_k \delta_k^{\mathsf{T}} \frac{\partial x(\lambda^*)}{\partial x} + \alpha_k \left( \frac{\partial x(\lambda^*)}{\partial x} \right) \tag{33}$$

$$= \left( \frac{\partial \alpha_k}{\partial \tilde{\Delta}} \right) \delta_k \delta_k^{\mathsf{T}} + \left( \alpha_k - \left( \frac{\partial \alpha_k}{\partial \tilde{\Delta}} \right) \Delta \delta_k \delta_k^{\mathsf{T}} \right) \left( \frac{\partial x(\lambda^*)}{\partial x} \right) \tag{34}$$

$$= \left( \frac{\partial \alpha_k}{\partial \tilde{\Delta}} \right) \delta_k \delta_k^{\mathsf{T}} + \left( \alpha_k - \left( \frac{\partial \alpha_k}{\partial \tilde{\Delta}} \right) \Delta \delta_k \delta_k^{\mathsf{T}} \right) \left( \lambda^* \Delta^{-1} I - \lambda^* \Delta^{-1} \delta_k \delta_k^{\mathsf{T}} + \delta_k \left( \frac{\partial \lambda^*}{\partial \delta_k} \right)^{\mathsf{T}} \left( \frac{\partial \delta_k}{\partial x} \right) \right) \tag{35}$$

$$= \left( \frac{\partial \alpha_k}{\partial \tilde{\Delta}} \right) \delta_k \delta_k^{\mathsf{T}} + \left( \alpha_k - \left( \frac{\partial \alpha_k}{\partial \tilde{\Delta}} \right) \Delta \delta_k \delta_k^{\mathsf{T}} \right) \left( \lambda^* \Delta^{-1} I - \lambda^* \Delta^{-1} \delta_k \delta_k^{\mathsf{T}} + \Delta^{-1} \delta_k \left( \frac{\partial \lambda^*}{\partial \delta_k} \right)^{\mathsf{T}} \left( I - \delta_k \delta_k^{\mathsf{T}} \right) \right) \tag{36}$$

$$= \left( \frac{\partial \alpha_k}{\partial \tilde{\Delta}} \right) \delta_k \delta_k^{\mathsf{T}} + \left( \alpha_k - \left( \frac{\partial \alpha_k}{\partial \tilde{\Delta}} \right) \Delta \delta_k \delta_k^{\mathsf{T}} \right) \left( \lambda^* \Delta^{-1} I - \lambda^* \Delta^{-1} \delta_k \delta_k^{\mathsf{T}} + \Delta^{-1} \delta_k \left( \frac{\partial \lambda^*}{\partial \delta_k} \right)^{\mathsf{T}} - \Delta^{-1} \left( \left( \frac{\partial \lambda^*}{\partial \delta_k} \right)^{\mathsf{T}} \delta_k \right) \delta_k \delta_k^{\mathsf{T}} \right) \tag{37}$$

$$= \left( \frac{\partial \alpha_k}{\partial \tilde{\Delta}} \right) \delta_k \delta_k^{\mathsf{T}} + \alpha_k \lambda^* \Delta^{-1} I - \alpha_k \lambda^* \Delta^{-1} \delta_k \delta_k^{\mathsf{T}} + \alpha_k \Delta^{-1} \delta_k \left( \frac{\partial \lambda^*}{\partial \delta_k} \right)^{\mathsf{T}} - \alpha_k \Delta^{-1} \left( \left( \frac{\partial \lambda^*}{\partial \delta_k} \right)^{\mathsf{T}} \delta_k \right) \delta_k \delta_k^{\mathsf{T}} \tag{38}$$

$$- \left( \frac{\partial \alpha_k}{\partial \tilde{\Delta}} \right) \lambda^* \delta_k \delta_k^{\mathsf{T}} + \left( \frac{\partial \alpha_k}{\partial \tilde{\Delta}} \right) \lambda^* \left( \delta_k^{\mathsf{T}} \delta_k \right) \delta_k \delta_k^{\mathsf{T}} - \left( \frac{\partial \alpha_k}{\partial \tilde{\Delta}} \right) \left( \delta_k^{\mathsf{T}} \delta_k \right) \delta_k \left( \frac{\partial \lambda^*}{\partial \delta_k} \right)^{\mathsf{T}} \tag{39}$$

$$+ \left( \frac{\partial \alpha_k}{\partial \tilde{\Delta}} \right) \left( \delta_k^{\mathsf{T}} \delta_k \right) \left( \left( \frac{\partial \lambda^*}{\partial \delta_k} \right)^{\mathsf{T}} \delta_k \right) \delta_k \delta_k^{\mathsf{T}} \tag{40}$$

$$= \alpha_k \lambda^* \Delta^{-1} I + \left[ \alpha_k \Delta^{-1} - \left( \frac{\partial \alpha_k}{\partial \tilde{\Delta}} \right) \left( \delta_k^{\mathsf{T}} \delta_k \right) \right] \delta_k \left( \frac{\partial \lambda^*}{\partial \delta_k} \right)^{\mathsf{T}} \tag{41}$$

$$+ \left[ \left( \frac{\partial \alpha_k}{\partial \tilde{\Delta}} \right) - \alpha_k \lambda^* \Delta^{-1} - \alpha_k \Delta^{-1} \left( \left( \frac{\partial \lambda^*}{\partial \delta_k} \right)^{\mathsf{T}} \delta_k \right) - \lambda^* \left( \frac{\partial \alpha_k}{\partial \tilde{\Delta}} \right) \left( \delta_k^{\mathsf{T}} \delta_k - 1 \right) + \left( \frac{\partial \alpha_k}{\partial \tilde{\Delta}} \right) \left( \delta_k^{\mathsf{T}} \delta_k \right) \left( \left( \frac{\partial \lambda^*}{\partial \delta_k} \right)^{\mathsf{T}} \delta_k \right) \right] \delta_k \delta_k^{\mathsf{T}} \tag{42}$$

$$= \alpha_k \lambda^* \Delta^{-1} I + \left[ \alpha_k \Delta^{-1} - \left( \frac{\partial \alpha_k}{\partial \tilde{\Delta}} \right) \right] \delta_k \left( \frac{\partial \lambda^*}{\partial \delta_k} \right)^{\mathsf{T}} \tag{43}$$

$$+ \left[ \left( \frac{\partial \alpha_k}{\partial \tilde{\Delta}} \right) - \alpha_k \lambda^* \Delta^{-1} - \alpha_k \Delta^{-1} \left( \left( \frac{\partial \lambda^*}{\partial \delta_k} \right)^{\mathsf{T}} \delta_k \right) + \left( \frac{\partial \alpha_k}{\partial \tilde{\Delta}} \right) \left( \left( \frac{\partial \lambda^*}{\partial \delta_k} \right)^{\mathsf{T}} \delta_k \right) \right] \delta_k \delta_k^{\mathsf{T}} \tag{44}$$

Now that $\frac{\partial f_k(x)}{\partial x}$ is in the form of $cI + u_1 v_1^\mathsf{T} + u_2 v_2^\mathsf{T}$, where

$$c \triangleq \alpha_k(\tilde{\Delta})\lambda^* \Delta^{-1} \tag{45}$$

$$u_1 \triangleq \left[ \alpha_k(\tilde{\Delta})\Delta^{-1} - \frac{\partial \alpha_k(\tilde{\Delta})}{\partial \tilde{\Delta}} \right] \boldsymbol{\delta_k}(\boldsymbol{x}) \tag{46}$$

$$v_1 \triangleq \frac{\partial \lambda^*}{\partial \boldsymbol{\delta_k}(\boldsymbol{x})} \tag{47}$$

$$u_2 \triangleq \left[ \left( \frac{\partial \alpha_k(\tilde{\Delta})}{\partial \tilde{\Delta}} \right) - \alpha_k(\tilde{\Delta})\lambda^* \Delta^{-1} - \alpha_k(\tilde{\Delta})\Delta^{-1} \left( \left( \frac{\partial \lambda^*}{\partial \boldsymbol{\delta_k}(\boldsymbol{x})} \right)^\mathsf{T} \boldsymbol{\delta_k}(\boldsymbol{x}) \right) + \left( \frac{\partial \alpha_k(\tilde{\Delta})}{\partial \tilde{\Delta}} \right) \left( \left( \frac{\partial \lambda^*}{\partial \boldsymbol{\delta_k}(\boldsymbol{x})} \right)^\mathsf{T} \boldsymbol{\delta_k}(\boldsymbol{x}) \right) \right] \boldsymbol{\delta_k}(\boldsymbol{x}) \tag{48}$$

$$v_2 \triangleq \boldsymbol{\delta_k}(\boldsymbol{x}) \tag{49}$$

Applying the matrix determinant lemma twice, we can show that

$$\det(cI + u_1 v_1^\mathsf{T} + u_2 v_2^\mathsf{T}) = (1 + v_2^\mathsf{T}(cI + u_1 v_1^\mathsf{T})^{-1} u_2) \det(cI + u_1 v_1^\mathsf{T}) \tag{50}$$

$$= \left[ \left( 1 + c^{-1} v_2^\mathsf{T} u_2 - \frac{c^{-2}}{1 + c^{-1} v_1^\mathsf{T} u_1} \right) v_2^\mathsf{T} u_1 v_1^\mathsf{T} u_2 \right] (1 + c^{-1} v_1^\mathsf{T} u_1) c^D \tag{51}$$

We simplify this by defining the scaled dot products,

$$w_{ij} \triangleq c^{-1} v_i^\mathsf{T} u_j, \text{ for } i, j \in \{1, 2\}. \tag{52}$$

Then

$$\log \left| \det \frac{\partial f_k(x)}{\partial x} \right| = \log |1 + w_{11}| + \log \left| 1 + w_{22} - \frac{w_{12} w_{21}}{1 + w_{11}} \right| + D \log c \tag{53}$$

$\square$

Intermediate steps above used the following gradient identities.

$$\begin{aligned}
\frac{\partial \boldsymbol{\delta_k}(\boldsymbol{x})}{\partial x} &= \frac{\partial}{\partial x}(x - x_k)\Delta^{-1} \\
&= \Delta^{-1} I + (x - x_k) \left( \frac{\partial}{\partial x} \left( \Delta^2 \right)^{-\frac{1}{2}} \right)^\mathsf{T} \\
&= \Delta^{-1} I + (x - x_k) \left( \frac{-1}{2\Delta^3} \right) \left( \frac{\partial \Delta^2}{\partial x} \right)^\mathsf{T} \\
&= \Delta^{-1} I + (x - x_k) \left( \frac{-1}{2\Delta^3} \right) 2 \left( x - x_k \right)^\mathsf{T} \\
&= \Delta^{-1} \left( I - \boldsymbol{\delta_k}(\boldsymbol{x})\boldsymbol{\delta_k}(\boldsymbol{x})^\mathsf{T} \right)
\end{aligned} \tag{54}$$

$$\frac{\partial \Delta}{\partial x} = \frac{\partial \Delta}{\partial x} = \frac{\partial}{\partial x} \left( \Delta^2 \right)^{\frac{1}{2}} = \frac{1}{\Delta}(x - x_k) = \boldsymbol{\delta_k}(\boldsymbol{x}) \tag{55}$$

**Log determinant of** $\frac{\partial f_k^{-1}(z)}{\partial z}$. We can also use Proposition 3 to compute the log determinant of the inverse transform without needing to recompute $f_k(x)$. The only difference is a sign: $\log \left| \det \frac{\partial f_k^{-1}(z)}{\partial z} \right| = -\log \left| \det \frac{\partial f_k(x)}{\partial x} \right|$. The required quantities, $\Delta$, $\boldsymbol{x}(\lambda^*)$, and $\boldsymbol{\delta_k}(\boldsymbol{x})$, are readily available after computing $x = f_k^{-1}(z)$. The gradients with respect to quantities of $x$ can be expressed using gradients with respect to quantities of $z$,

$$\frac{\partial \alpha_k(\Delta)}{\partial \Delta} = \left( \frac{\partial \alpha_k^{-1}(\tilde{\alpha})}{\partial \tilde{\alpha}} \right)^{-1} \quad \text{and} \quad \frac{\partial \lambda^*}{\partial \boldsymbol{\delta_k}(\boldsymbol{x})} = \frac{\partial \lambda^*}{\partial \boldsymbol{\delta_k}(\boldsymbol{z})}, \tag{56}$$

which are accessible through automatic differentiation.

**Proposition 1.** *The mapping $f_k : \mathbb{R}^D \to V_k$ as defined in the 2-step procedure is a homeomorphism.*

*Proof.* Let $x \in \mathbb{R}^D$ and $x_k$ be a given anchor point corresponding to a Voronoi cell $V_k$. If $x \neq x_k$, then $x$ is uniquely represented by the tuple $(\Delta, \delta)$ where $\Delta = \|x - x_k\|$ and $\delta = \frac{x - x_k}{\|x - x_k\|}$, since $x = x_k + \Delta\delta$. Since $\delta$ uniques defines the ray $\{x_k + \lambda\delta; \lambda > 0\}$ and $x(\lambda^*)$, then because $\alpha_k$ in Equation (8) is a bijection in $\Delta$, $f_k$ is a bijection. For $x \neq x_k$, the continuity of $f_k$ follows from Rudin et al. [39, Theorem 4.7] since $\Delta$ and $x(\lambda^*)$ are continuous in $x$ and $\alpha_k$ is continuous in $\Delta$. Then since $f_k(x) \to x_k$ as $x \to x_k$ from all directions, this justifies the choice of setting $f_k(x) = x_k$ when $x = x_k$. Finally, by the invariance of domain theorem, since $V_k$ is an open set in $\mathbb{R}^D$, $f_k$ is an open map and the inverse $f_k^{-1}$ is continuous, and we can conclude $f$ is a homeomorphism between $\mathbb{R}^D$ and $V_k$. □

**Proposition 2.** *If $p_x(x)$ is continuous, then the transformed density $p_z(f_k(x))$ is continuous almost everywhere.*

*Proof.* See proof of Proposition 3 below for the form of the Jacobian. For the case where $x \neq x_k$, all quantities in Equations (22) to (29) are continuous with respect to $x$. Hence the Jacobian $\frac{\partial f_k(x)}{\partial x}$ is continuous, and since it is always full-rank, then the composition $\left|\det \frac{\partial f_k(x)}{\partial x}\right|$ is continuous [39, Theorem 4.7] and so is the product $p_x(x) \left|\det \frac{\partial f_k(x)}{\partial x}\right|$ [39, Theorem 4.9]. □

**Proposition 4.** *For any distribution $p(k)$ with support over $\{1, \ldots, K\}$, define the mixture distribution,*

$$p(x) = \sum_{k=1}^{K} p(x|k)p(k), \tag{19}$$

*where $p(\cdot|k)$ is the distribution mapped onto the Voronoi cell $V_k$ (i.e. Equation 10) from a density that is continuous. Then the density function of the mixture is continuous a.e.*

*Proof.* Proposition 2 ensures the distribution is continuous a.e. within each Voronoi cell. It's straightforward to see that the density of $p_z(f_k(x))$ approaches zero as $f_k(x)$ approaches the boundaries of $V_k$ for any properly normalized distribution $p_x$. □

## G  DATA SETS

### G.1  UCI DATA SETS

The main preprocessing we did was to (i) remove the "label" attribute from each data set, and (ii) remove attributes that only ever take on one value. Apart from this, the USCensus90 dataset contains a unique identifier for each row, which was removed. Descriptions for all dataset are below.

**Connect4** [webpage]   This dataset contains all legal 8-ply positions in the game of Connect Four in which neither player has won yet, and in which the next move is not forced. The original task was to predict which player would win, which has been removed during preprocessing. There are a total **42** discrete variables (one for each location on the board), each with **3** possible discrete values (taken by player 1, taken by player 2, blank). This data set was randomly split into $54045$ training examples, $6755$ validation examples, and $6757$ test examples.

**Forests** [webpage]   This dataset contains cartographic variables regarding forests including four wilderness areas located in the Roosevelt National Forest of northern Colorado. These areas represent forests with minimal human-caused disturbances. The original task was to predict the forest cover type, which has been removed during preprocessing. There are a total of **54** discrete variables, with **10** being the highest number of discrete values. This data set was randomly split into $464809$ training examples, $58101$ validation examples, and $58102$ test examples.

**Mushroom** [webpage]  This data set includes descriptions of hypothetical samples corresponding to 23 species of gilled mushrooms in the Agaricus and Lepiota Family. The original task was to predict whether each species is edible, which has been removed during preprocessing. There are a total of **21** discrete variables, with **12** being the highest number of discrete values. This data set was randomly split into 6499 training examples, 812 validation examples, and 813 test examples.

**Nursery** [webpage]  This data set contains attributes of applicants to nursery schools, during a period when there was excessive enrollment to these schools in Ljubljana, Slovenia, and the rejected applications frequently needed an objective explanation. All data have been completely anonymized. The original task was to predict whether an applicant would be recommended for acceptance by hierarchical decision model, which has been removed during preprocessing. There are a total of **8** discrete variables, with **5** being the highest number of discrete values. This data set was randomly split into 10367 training examples, 1296 validation examples, and 1297 test examples.

**PokerHands** [webpage]  This data set contains poker hands consisting of five playing cards drawn from a standard deck of 52. Each card is described using two attributes (suit and rank), for a total of 10 predictive attributes. There is one Class attribute that describes the "poker hand". The original task was to predict the poker hand class (pairs, full house, royal flush, etc.), which has been removed during preprocessing. There are a total of **10** discrete variables, with **13** being the highest number of discrete values. This data set was randomly split into 820008 training examples, 102501 validation examples, and 102501 test examples.

**USCensus90** [webpage]  This data set contains a portion of the data collected as part of the 1990 census in the United States, with the data completely anonymized. There are a total of **68** discrete variables, with **18** being the highest number of discrete values. This data set was randomly split into 2212456 training examples, 122914 validation examples, and 122915 test examples.

### G.2   ITEMSET DATA SETS

These data sets were taken from the Frequent Itemset Mining Dataset Repository [webpage]. Each row is interpreted as a set of items with no emphasis on the ordering of items.

**Retail** [3]  This data set contains anonymized retail market basket data from an anonymous Belgian retail store. We first removed rows with less than 4 items, then randomly sampled a subset of 4 items for every row. Items that appear in less 300 rows were dropped from the data set. The final preprocessed data set contains **765** distinct items. This data set was randomly split into 24280 training examples, 3035 validation examples, and 3036 test examples.

**Accidents** [11]  This data set contains contains anonymized traffic accident data. Data on traffic accidents are obtained from the National Institute of Statistics (NIS) for the region of Flanders (Belgium) for the period 1991-2000. We first removed rows with less than 4 items, then randomly sampled a subset of 4 items for every row. This subsampling occurred 10 times if a row has 16 or more items, 5 times if the row has 8 to 15 items, and once if the row has 4-7 items. Items that appear in less 300 rows were dropped from the data set. The final preprocessed data set contains **213** distinct items. This data set was randomly split into 270129 training examples, 33766 validation examples, and 33767 test examples.

## H   EXPERIMENTAL DETAILS

**2D synthetic data sets**  Continuous data sets were quantized into 91 bins for each coordinate. For Voronoi dequantization, we dequantized each coordinate into an embedding space of 2 dimensions, with 91 Voronoi cells. The dequantization model is parameterized by 4 layers of coupling blocks, each with a 2 hidden layer MLP with 256 hidden units each, where the Swish activation function was used [37]. The flow model is similarly parameterized but with 16 layers of coupling blocks. Each block alternated between 4 different partitioning schemes: maksing out the first half, masking out the second half, masking out the odd indices, and masking out the even indices. We trained with the `Adam` optimizer [21] with a learning rate of `1e-3`.

**UCI data sets**  For Voronoi dequantization, we dequantized each coordinate into an embedding space of 4 or 6 dimensions, with the number of Voronoi cells set to the highest number of discrete values over all discrete variables. The dequantization model is parameterized by 4 layers of coupling blocks, each with a 2 hidden layer MLP with 256, 512, or 1024 hidden units each, where the Swish activation function was used [37]. The flow model is similarly parameterized but with 16 or 32 layers of coupling blocks. Each block alternated between 4 different partitioning schemes: maksing out the first half, masking out the second half, masking out the odd indices, and masking out the even indices. We trained with the `Adam` optimizer [21] with a learning rate sweep over $\{$`1e-3, 5e-4, 1e-4`$\}$.

**Itemset data sets**  For Voronoi dequantization, we dequantized each coordinate into an embedding space of 6 dimensions, with the number of Voronoi cells set to the number of items in the data set. We used a continuous normalizing flow (CNF) with the ordinary differential equation (ODE) defined using a Transformer archiecture and a $L_2$-distance based multihead attention layer [20] and the GeLU activation function [14]. No positional embeddings were provided to the model to ensure the model is equivariant to permutations. We composed 12 CNF layers, each defined using a Transformer model that has 2 or 3 layers of alternating multihead attention and fully connected residual connections. To solve the ODE and train our model, we used the `dopri5` solver from the torchdiffeq library [4] with `atol=rtol=1e-5`. We trained with the `AdamW` optimizer [29, 47] with a learning rate of `1e-3` and weight decay of `1e-6`. For the Voronoi dequantization, we set $D$=6 for both data sets, though it may be possible to improve performance by tuning $D$.

**Character-level language modeling**  We used the provided hyperparameters from the open source repository [URL]. Some parts of code had to be adapted for our usage, but model architecture and optimizer remained the same.

