# OpenReview forum: "Semi-Discrete Normalizing Flows through Differentiable Voronoi Tessellation"
_ICLR.cc/2022/Workshop/DGM4HSD — ICLR 2022 DGM4HSD workshop Oral_

### Official Review · Reviewer_RaSc · 2022-03-21
**Exciting and novel method with great potential**

**Rating:** 10
**Confidence:** 5

**Review:**

**Summary**

The submission introduces a trainable Voronoi tesselation which combined with normalizing flows allows training flexible disjoint mixture densities for continuous data. This is realized by a non-smooth projection of the real space onto a compact polytope having a tractable exact Jacobian determinant as well as tractable inverses. The mapping can be efficiently computed as the result of a linear program. To avoid a continuity problem of unbounded Voronoi cells compact support of the target density is assumed.

The submission furthermore shows that this approach can be seen as a generalization of many recently introduced dequantization tricks. The method is extensively evaluated on a range of meaningful benchmark tasks, comprising density estimation on synthetic quantized 2D data, four discrete UCI datasets, permutation-invariant item set data, and MNIST and character-level language modeling. In all cases, the method either outperforms or closely matches strong baselines.

**Discussion**

The work aligns very well with the workshop and proposes a novel and very interesting method with plenty of opportunities for future applications.

The method is well-motivated and all necessary mathematical details (derivation of the Jacobian determinant, gradients, inverses, etc.) are worked out well without clutter. It is very clearly written and easy to read for a researcher with knowledge in the field.

The experiments show that the method works competitively on a variety of hard benchmarks even outperforming state-of-the-art competing methods. The experiments are well-chosen and all the details to reproduce them are given in the main text or the appendix.

The authors have spent a significant effort on explaining limitations and discussing the computational costs of the method as well as opportunities for future improvements. They furthermore give a holistic view of how this work relates to prior work.

I can not come up with any ideas for improving the paper, except for more extensive experiments or possibly other applications of the method (which probably go beyond the scope of a workshop paper). E.g., it would be interesting to see how it could help for improving density estimation on semi-continuous data like graphs with continuous properties on nodes and/or edges.

**Recommendation**

I strongly recommend accepting this work at the workshop.

---

### Official Review · Reviewer_P5Wa · 2022-03-24
**Very high quality paper for a workshop, clear accept**

**Rating:** 8
**Confidence:** 4

**Review:**

The authors propose to use Voronoi tesselations to cleanly and intuitively model discrete data using normalizing flow methods. This approach provides a more natural method for dequantization; a simpler, less data-dependent approach is typically used instead when modelling discrete data with normalizing flows. This work also presents a new method to add disjoint mixture modelling to pre-existing models, resulting in gains in expressiveness at an insignificant computation cost. Altogether, this paper provides an interesting perspective on discrete modelling with normalizing flows, along with some compelling experiments. __I highly recommend this paper for publication at the workshop.__

I would like to bring up some questions/concerns I have though which I do not believe were adequately addressed in either the main paper or appendix (although I did not thoroughly read the appendix, so I apologize if these are addressed there):
1. I am not totally convinced that the approach in its current form truly does scale to higher dimensions. I am worried about partitioning a very high-dimensional domain because of the curse of dimensionality. Some clarity here would be appreciated.
    - Worth noting that the results in __Table 1__ would at first glance appear to assuage my concerns, but I am left wondering if the increases in log probability are simply a result of the domains being shrunk by the tessellation: for example, the uniform distribution on [0,1] has a higher average log-likelihood than the uniform distribution on [-1,1]. Is a similar effect happening here? Can the authors provide some clarity on this point?
2. More of a thought experiment - how would this approach interact with something like [Local Bits Back Coding](https://arxiv.org/abs/1905.08500) (worth noting that I am NOT an author of this work), which itself requires a quantization of continuous data to work correctly? Could we combine the approaches to achieve better compression?

Some small typos on P3:
- There are extra brackets in (7)
- There are two "1."s giving instructions. Was this intentional?

---

### Decision · Program_Chairs · 2022-03-26

Accept (Oral)